# The Consequences of Water Interactions with Nitrogen-Containing Carbonaceous Quantum Dots—The Mechanistic Studies

**DOI:** 10.3390/ijms232214292

**Published:** 2022-11-18

**Authors:** Marek Wiśniewski

**Affiliations:** Physicochemistry of Carbon Materials Research Group, Faculty of Chemistry, Nicolaus Copernicus University in Toruń, Gagarina 7, 87-100 Toruń, Poland; marekw@umk.pl

**Keywords:** carbon quantum dots, H_2_O dissociative adsorption, adsorption enthalpy, diffusive layer, Stern layer, Eigen–Zundel complexes

## Abstract

Despite the importance of quantum dots in a wide range of biological, chemical, and physical processes, the structure of the molecular layers surrounding their surface in solution remains unknown. Thus, knowledge about the interaction mechanism of Nitrogen enriched Carbonaceous Quantum Dots’ (N-CQDs) surface with water—their natural environment—is highly desirable. A diffusive and Stern layer over the N-CQDs, characterized in situ, reveals the presence of anionic water clusters [OH(H_2_O)_n_]^−^. Their existence explains new observations: (i) the unexpectedly low adsorption enthalpy (ΔH_ads_) in a pressure range below 0.1 p/p_s_, and ΔH_ads_ being as high as 190 kJ/mol at 0.11 p/p_s_; (ii) the presence of a “conductive window” isolating nature—at p/p_s_ below 0.45—connected to the formation of smaller clusters and increasing conductivity above 0.45 p/p_s_, (iii) Stern layer stability; and (iv) superhydrophilic properties of the tested material. These observables are the consequences of H_2_O dissociative adsorption on N-containing basic centers. The additional direct application of surfaces formed by N-CQDs spraying is the possibility of creating antistatic, antifogging, bio-friendly coatings.

## 1. Introduction

Due to similarities to their inorganic cognates, Carbon Quantum Dots (CQDs) have attracted significant and growing interest in scientific and technological branches. In this sense, CQDs possess powerful and tunable fluorescence, excellent photostability, electrochemiluminescence, exceptional multi-photon excitation (up-conversion), and other properties enabling their applications in optronics, sensors, and catalytic applications [1,2].

Their light-absorption properties have been exploited as means to enhance photocatalysis. The CQDs were used for the selective photo-oxidation of benzyl alcohol to benzaldehyde (under vis-light), and the photocatalytic generation of H_2_ (under UV-light).

Two distinct approaches to synthesizing CQDs have been proposed: the graphitized CQDs—synthesized via “cutting” the graphitized precursor, and the amorphous CQD—usually obtained via the hydrothermal treatment of carbon-rich molecular precursors. The carbon source for the latter method can be inter alia carbohydrates [3], proteins [4,5], soy milk [6], and polyacrylamide [7]. Nitrogen-rich precursors, such as amino acids, peptides, or N-rich proteins, allow for obtaining specific, N-enriched CQDs. The indisputable advantages of N-CQDs are their extremely low toxicity and outstanding biodegradability. Thanks to this attribute, their biomedicine applications are quite natural, e.g., as effective carriers for drug delivery and biological imaging [8,9,10,11]. However, other applications, e.g., in electronics, sensors, and batteries, are widely investigated [12].

None of the sabove-mentioned features would be possible if CQDs would not be perfectly dispersed in water, which requires strong interaction with the solvent. The subject was not considered until now.

Thus, it is crucial to analyze CQDs–water interactions, which are affected by the rich surface chemistry of N-CQDs causing H_2_O dissociation. Consequently, Eigen–Zundel (E–Z) water complexes are expected to be formed [13,14,15,16], and careful FTIR analysis can be irreplaceable here.

On the other hand, to characterize E–Z water complexes, very useful in our divagation is the concept of hardness or softness, introduced by Pearson [17,18] for studying acid–base chemical reactions of the type A + :B → A:B, where A, the acceptor of electrons, is the acid, and B, the donor of electrons, is the base. Acids and bases are thus classified as hard or soft, based on the polarizability and size of the system. The theory has been used successfully to interpret the adsorption of metal or chlorine ions or other adsorbates onto activated carbons [19,20].

The description of the properties of water clusters treated as models using the HSAB theory is not very popular [21]. Moreover, to the best of our knowledge, it is novel for water clusters. Although by taking into account molecular high polarizability, expressed as relatively accessible charge movement (in the sense of ionization potential and electron affinity values), the high conductivity should be a natural feature of soft acids and bases.

Besides the above, some recently-appeared review articles [22,23,24,25,26] describe unique properties of different (inorganic, organic, as well as carbon) quantum dots; nevertheless, the interactions with their natural solvent—water are omitted. Thus, there is a great need to shed some light on the problem. To the best of the author’s knowledge, this is the first report showing the adsorption isotherm and enthalpy of water on the CQDs surface. Absolute novelty is also the explanation of the superhydrophilicity mechanism via a stable Stern layer.

## 2. Results and Discussion

It is known that antistatic agents being sprayed onto the surface of insulators can create a conductive film, while being exposed to water, which prevents the surface from becoming electrically charged and distributes the water droplets as a thin film with a 0° contact angle. The results from this simple experiment, by using N-CQDs (Figure 1), possess far-reaching consequences that have been studied and analyzed in this work.

Firstly, the superhydrophilic character of the tested material (Figure 1b) is evident. The following observation shows the formed layer’s stability, which means that the N-CQDs film does not require frequent spraying when used as an antifogging agent. Moreover, by combining our recent results, proving that they are highly biocompatible [8,27], the N-CQDs are not harmful to the environment. The material can easily be used as anti-fog drops, e.g., in diving masks.

The next natural problem is connected with conductivity: a fast rise and drop to a zero value. The results presented in Figure 2 explain this phenomenon. The surface functionalities of the dots behaving as acids or bases cause the adsorbed water film to become conductive, starting from a certain pressure. This, in turn, reduces the resistance of the insulator and results in a decreased local electrical charge. Additionally, higher humidity causes more significant moisture on the surface of the sample surface. The simple acid–base protolysis processes and their high kinetic energy are the reason for: the nanolayer’s fast electrical response, complete reversibility, linearity vs. RH, and stability for the N-CQDs material (Figure 2).

The above results prove the strong interaction of N-CQDs with water molecules. Thus, they are the key to explaining how much water can be adsorbed on the N-CQDs surface. The adsorption isotherm was determined, and the results are presented in Figure 3.

When water is adsorbed on highly hydrophilic surfaces, the D’Arcy and Watt (DW) adsorption isotherm model (Appendix A) can be considered [28]. This simple approximation assumes that water adsorption occurs independently on stronger—high-energy, and weaker—low-energy, sites. The former primary binding centers consist here mainly of superhydrophilic groups. Their calculated number is as high as 8.95 mmol/g (Appendix A). The value of the Langmuir constant (K_L_), which is the parameter defining the energy of the process, also reaches a very high value of ca. 296. Such tremendous values of the Langmuir components of the DW model are responsible for the almost “squared-shape” of the isotherm in the low-pressure range, where adsorption is 160 mg/g (the a_mL_ value) at p/p_s_ = 0.05.

A further increase in the H_2_O pressure reveals the 2nd type of adsorption, isotherm, which is also suitable for fitting with the DW model. The value of parameter *c*—defined as the ratio of adsorption and desorption constants—*c* = 0.89, suggests highly-interacting sites. Note that the conductive layer is formed here, where the adsorption in the second layer (DS1) is at least as high as the a_mL_ value.

In the model, it is simplified so that there is only one type of secondary site on which, for the adsorbate water, two-or three-dimensional hydrogen-bonded clusters can begin to build up, even before all the primary sites are occupied (see inset in Figure 3a). Thus, the estimated values of a_0_ and c guarantee very high adsorption capacity, ca. 2 g(H_2_O)/g(N-CQDs) at 100% humidity, as well as good ionic conductivity of the close-to-the-surface nanolayer. Moreover, due to the lack of a hysteresis loop, one can expect that this diffusive layer will be quite large, and their formation is fully reversible. Our recent results showed that, calculated from Perrin’s equation, the radius of the effective redox zone around the N-CQDs surface is 5-nm-wide [8].

On the contrary, the measured differential heat of adsorption values in the low-pressure range is deficient (Figure 3b). Such a situation seems to be unusual, taking into account the number of high-energy sites. To explain this phenomenon, one has to consider that the total effect is observed. The algebraic sums of the energetically opposed reactions in the Stern layer are measured as values of q^diff^. The first natural candidate is the H_2_O auto dissociation with ΔH = +57.3 kJ/mol [21]. The competition reaction is amine protonation (see further sections) for which ΔH = −55–60 kJ/mol [29]. The assumption explains the double signals (endo- and exothermal) shown in Appendix A, which are clearly observed in the low-pressure range.

Thus, starting from only these two processes, the initial values of |q^diff^| should be close to zero. An increase in the H_2_O pressure causes the increase in the H_2_O adsorption contribution, and thus the increase in the |q^diff^| value up to maximal value, ca. 200 kJ/mol at p/p_s_ = 0.11, which corresponds to adsorption of 9.15 mmol(H_2_O)/g(N-CQDs). Further, a rise in H_2_O pressure causes a decrease in the measured heat of adsorption, down to ca. 25 kJ/mol, and a subsequent increase in the value of H_2_O condensation (44 kJ/mol). The observed minimum is connected closely to the formation of a diffusive layer (see next section) and the increasing participation of the DS1 part in the DW model.

Water is known to have a high affinity toward each polar site. Subsequently, those molecules adsorbed on the primary sites can become secondary ones, and adsorption on them occurs following the original Dubinin–Serpinsky mechanism. Thus, the total heat of adsorption approaches the condensation value and formation of the conductive layer. The desorption and adsorption branches of the isotherm are fully covered, similarly to the |q^diff^| curves, in almost the whole pressure range. The exception is the very low-pressure range below 0.02 p/p_s_, where hysteresis starts to be visible. One can postulate that the difference is the Stern layer formation enthalpy net effect (ca. −1.4 kJ/mol).

The spontaneity of the H_2_O adsorption on N-CQDs is confirmed by negative values of Gibbs free energy (ΔG) in the whole range.

In the low adsorption range (up to ca. 9 mmol/g), the ΔG is dominated by an entropic factor (Figure 3c), whose positive value confirms that, after the H_2_O dissociation, the OH^−^ ions are continually moving throughout, as they are attracted and repulsed by the polar ends of the water molecules. Therefore, the deep diffuse layer is more disordered, having higher entropy.

The decrease of ΔS and ΔH in the Langmuir region of adsorption isotherm suggests that additional exothermic processes occur. The decrease of ΔS is combined with the formation of [OH(H_2_O)_n_]^−^ clusters [13,30,31]. Their adsorption on the primary sites is strongly exothermic. An increase in H_2_O pressure causes a slightly higher enthalpy contribution to the ΔG value. As one can expect, the ΔG asymptotically tends to zero, which means that, even at 100% humidity, the process of H_2_O adsorption on the N-CQDs can reach equilibrium.

Surface chemistry is the key factor to understanding the observed phenomena: high adsorption capacity and surface ionic conductivity, and the alkaline (pH = 9.5) character of the N-CQDs solution. The design in our lab experiment, called the “evaporating droplet ATR technique”, is proposed to perform this.

The method enables the observation and characterization of selected coatings, i.e., bulk, diffusive, or Stern layer above the surface of the tested materials via the controllable removal of water deposits and thus selective uncovering of the hidden layers. Its simplicity is based on the slow evaporation of the droplet placed on the ATR crystal (Figure 4a). The needle valve regulates the diminishing pressure, enabling registration of the spectral changes. Figure 4b reveals how important and valuable this technique is. While the IR spectra of quantum dots are usually recorded without the solution’s removal, strong H_2_O absorption disturbs the appropriate analysis by shielding the dispersed phase [32,33]. The proposed technique naturally reveals the particular layers above the CQDs surface.

We need to look closer at the spectral changes in the most interesting, i.e., diffusive layer. The results are collected in Figure 4c,d. It is important to note that water does not evaporate totally under these conditions. The effect is visible as wide and mutually overlapped IR bands in the ν(OH), and in the wagging vibrations region (800–300 cm^−1^), respectively [31]. The presence of the 3352, 3184 bands and the ones below 2800 cm^−1^ prove the existence of the Eigen-like water complexes [14,15,16,30,34] and protonated surface NH_x_ structures at 3050 cm^−1^ [35]. The wide signals at 2950 and 2878 cm^−1^ cannot be uncritically and unequivocally attributed to CH stretching vibrations. To prove this statement, the isobar in an H_2_O vapor environment (25–125 °C under 30 hPa) was taken (Appendix A). The gradual decrease in the intensity of the bands at 3470, 3245, 2605, and 2081 cm^−1^ with the temperature rise allows their attribution to the H_2_O-structures. Evaporating from the CQDs surface, the H_2_O molecules uncover the CH-containing structures. Note that their presence is totally hidden when water covers the CQDs.

Further analysis of the water clusters, aiming at shifting the 2600 and 2100 cm^−1^ bands explanation, was performed by ab initio calculations (Appendix A). The simulated infrared spectra of optimized [OH(H_2_O)_n_]^−^ (n = 1–17) structures show the presence of the IR bands, not only in the 3800–3000 cm^−1^ region but also in the 2000–3000 cm^−1^. The latter band’s location seemingly depends on the size of the Eigen structure, which was the reason for additional analysis—the IR band position vs. the OH-bond length (Appendix A).

The position of the stretching H_2_O bands is determined by the R(OH) length. To conclude, the appearance of the IR bands at ca. 2605 and 2081 cm^−1^ (Appendix A) means that, respectively, the 1.05 and 1.14 Å bond lengths are present in the water cluster.

The other added value of the presented experiment is the observation of IR-band intensities increasing and shifting with the evaporation time, especially in the fingerprint region (1800–1000 cm^−1^). While the intensity rise is connected with the decreasing bulk-H_2_O shielding effect, the shifting is via an interaction with the water molecules. If observed, the latter result is red-targeted, meaning that water clusters become smaller during evaporation, thus easily interacting with the surface functionalities.

Interestingly, there are no clear IR bands of the surface C=O structures [36]. One of the most probable reasons is their protonation and red-shifting down to ca. 1640 cm^−1^. One can assume that 1637 and 1396 cm^−1^ are attributed to I and III amide bands ν(C=O) and ν(C–N) vibrations. The II-amide is overlapped with the surface-protonated amine-bending vibrations, for which 1571 and 1105 cm^−1^ can be identified as NH_x_^+^-bending and C-N-stretching. This is evidence of the H-bonding of the C=O surface structures’ appearance.

The utility of the HSAB theory to describe the reactivity of different types of [OH(H_2_O)_n_]^−^ (n = 1–17) clusters can help to explain these observables. First, let’s analyze the theoretical reactivity indices: the ionization potential (IP), and electron affinity (EA) of the modeled structures (Appendix A). Due to the anionic form of the considered systems, the IP value is lower than the EA for small clusters. Consequently, the global hardness (η) value is highly negative—thus represents the insulating character of the H_2_O sets. When n increases, decreasing their differences causes η to become zero (Figure 5), and electronic conductivity appears. For n ≥ 11, the IP energy becomes higher than the EA, resulting in the total hardness becoming positive and reaching a plateau from n = 15. For the latter structures, the ionic conductivity becomes meaningful.

This simple analysis needs further confirmation but explains why the conductivity of the N-CQDs surface starts from ca. 45 % and increases with the rise in the RH.

## 3. Materials and Methods

### 3.1. The N-CQDs Synthesis

Nitrogen-enriched carbon quantum dots (N-CQDs) were prepared via the hydrothermal carbonization of gelatin at 200 °C for 6 h, under 19 MPa. After cooling down to room temperature, the obtained brown solution (N-CQDs suspension) was ready for further examination. We have compared the fluorescence spectra for an as-obtained sample and two others subjected to further purification processes (centrifugation at 15,000× *g* for 30 min, and dialysis).

### 3.2. N-CQDs Physicochemical Characterization

The full physicochemical characterization was performed and described recently [8,27]. These analyses concern: the effect of purification, and the elemental analysis of the prepared N-CQDs samples performed by the energy dispersive X-ray method (EDX), AFM, and HRTEM.

The equipment used for the EDX analysis was the LEO Electron Microscopy Ltd., Cambridge, England, model 1430 VP.

The High-Resolution Transmission Electron Microscopy (HRTEM) images (Appendix A) were taken using a transmission electron microscope, F20X-TWIN (FEI-Tecnai), operated at 200 kV. The drop of sample solution was placed on a Cu grid coated with an ultrathin amorphous carbon film and then dried under ambient conditions. 

The morphology of the N-CQDs dispersed on Si wafers was analyzed at room temperature in the air using a microscope with a scanning SPM probe of the NanoScope MultiMode type (Veeco Metrology, Inc., Santa Barbara, CA, USA), which operated in a tapping mode. 

The fluorescence spectra were measured with a fluorescence spectrometer, RF-5001PC (Shimadzu, Japan). The excitation maximum was experimentally established at 360 nm. The UV-vis spectra of the N-CQDs were acquired using a Jasco 660 spectrometer in a range 200–800 nm.

Presented here, the SEM pictures show the N-CQDs immobilized on glass beads (Appendix A).

### 3.3. ATR FTIR Analysis

Fourier-transform infrared (FTIR) spectroscopy data were acquired by using a Vertex V70 (Bruker Optic), with ATR-mode techniques (single reflection using diamond crystal), in the frequency range 6000–15 cm^−1^.

### 3.4. Calorimetric Analysis

The water vapor adsorption isotherms were measured volumetrically at 298 K and the differential enthalpy of adsorption (q^diff^) was determined using a Tian–Calvet isothermal microcalorimeter constructed in our laboratory [37]. The related molar entropy changes of the adsorbed molecules (ΔS) were calculated using the basic thermodynamic relations (Appendix A); where R is the gas constant, and p and p_s_ are the equilibrium and saturated vapor pressures, respectively.

### 3.5. Ab Initio Studies

All the calculations (at B3LYP levels of theory using 6–311G(d,p) basis sets) were performed using geometries optimized for the parent compounds, as shown in Appendix A. These calculations were performed using the Gaussian ‘09 program [38]. For the calculation of the ionization potentials and electron affinities for the systems with the same geometry, with N, the N − 1 and N + 1 electrons were calculated (note that Koopmans’ theorem in terms of HOMO and LUMO energies is not satisfied in this case Appendix A). Thus, the reactivity of these model layers is discussed, on the basis of the global reactivity parameters, as the IP, EA, and global hardness (η).

## 4. Conclusions

The general conclusion from this study is that one cannot consider the CQDs’ surface properties without its surroundings—formed by water, which was experimentally proved here for the first time.

From the analysis of the mechanism of the water interactions with the N-CQDs surface in the whole pressure range presented in this study, one can conclude that:(i)the dissociative H_2_O chemisorption on superhydrophilic surfaces, for which 0° WCA, occurs in a low-pressure range;(ii)effective protolysis causes the formation of the [OH(H_2_O)_n_]^−^ water clusters arising during this process;(iii)their size (n) determines the conductivity;(iv)formed thus, Eigen-like water complexes appear in the diffusive and Stern layers and create a dense, stable, and nonconductive layer close to the N-CQDs surface;(v)proposed here, the “evaporating droplet ATR technique” is perfect for the characterization of materials soluble in water.

## Figures and Tables

**Figure 1 ijms-23-14292-f001:**
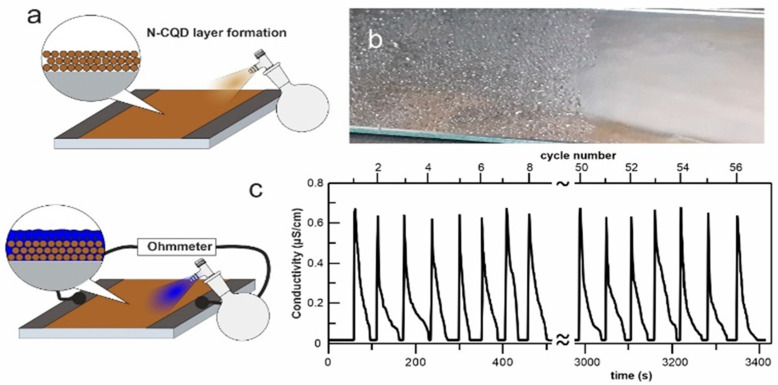
Conductive (**a**) and wetting (**b**) properties of antistatic nanolayer formed from N-CQDs. Good adhesion of the superhydrophilic nanomaterial causes complete reversibility of the sprayed layer tests (**c**).

**Figure 2 ijms-23-14292-f002:**
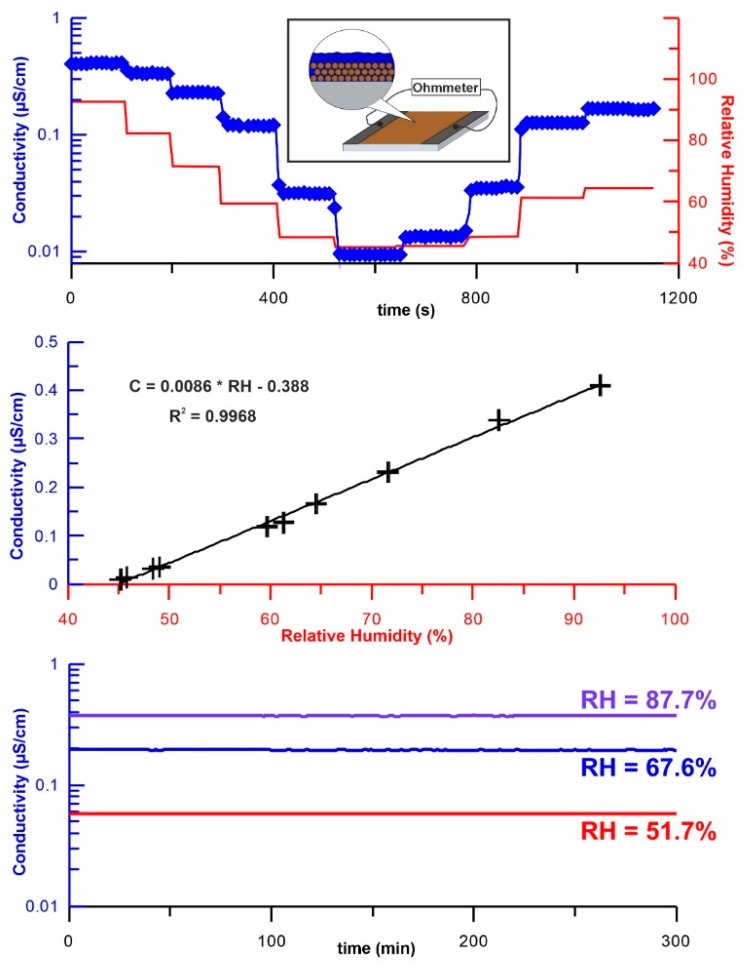
The conductive properties of nanolayer formed from N-CQDs. Acid–base reactions are responsible for kinetic energy, linearity vs. RH, and stability.

**Figure 3 ijms-23-14292-f003:**
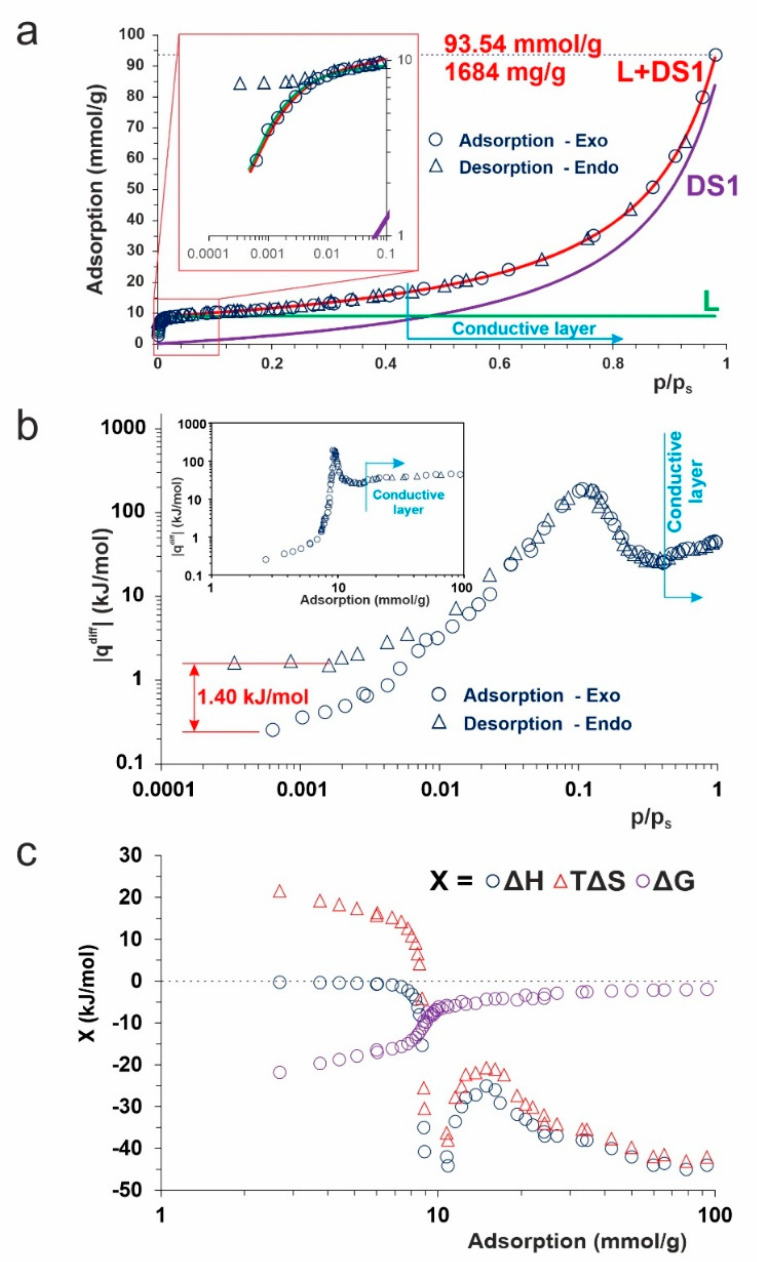
Adsorption of water on N-CQDs: (**a**): adsorption isotherm—red line—the DW model (Appendix A), green and purple lines—its Langmuir (L) and Dubinin–Sierpinsky (DS1) components, respectively; inset—the adsorption isotherm in the low-pressure region. The extremely high adsorption capacity reaches 1.7 g(H_2_O)/g(N-CQDs) at p/p_s_ = 0.98, while the used model predicted almost 2.0 g(H_2_O)/g(N-CQDs) under 100% humidity. (**b**): experimental data of differential heat of adsorption as a function of relative pressure and adsorption (inset). The calculated difference between adsorption and desorption branches means the energy of Stern layer formation; (**c**): changes in the thermodynamic parameters while N-CQDs are exposed to H_2_O vapor.

**Figure 4 ijms-23-14292-f004:**
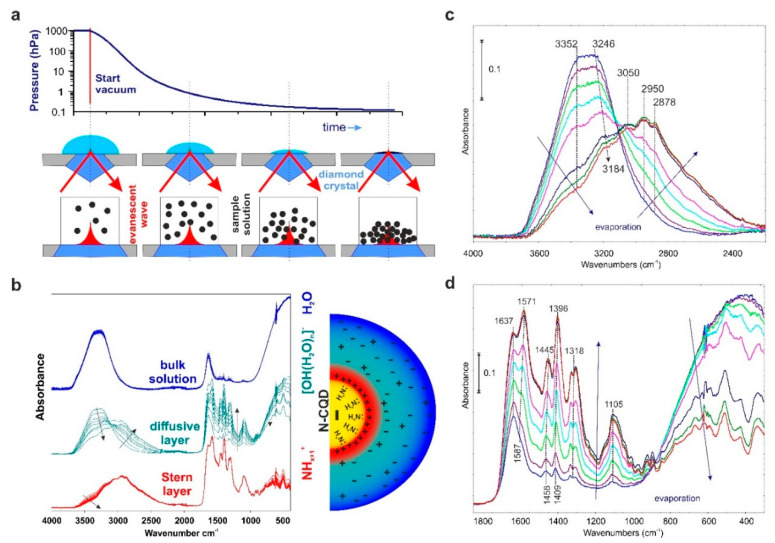
The “evaporating droplet ATR” technique scheme: (**a**): schematic representation of experimental setup allowing the pressure control and thus H_2_O evaporation kinetic; (**b**): the resulting ATR spectra separated into three distinct layers: the bulk (blue), the diffusive (green), and Stern layers (red). Arrows denote spectral changes registered during the evaporation process. At the same time, the primary species responsible for each layer’s existence are picked up on the scheme; (**c**,**d**) show the time-dependent evolution of spectral changes (labeled with blue arrows) in the diffusive layer. The spectra were recorded every 20 s.

**Figure 5 ijms-23-14292-f005:**
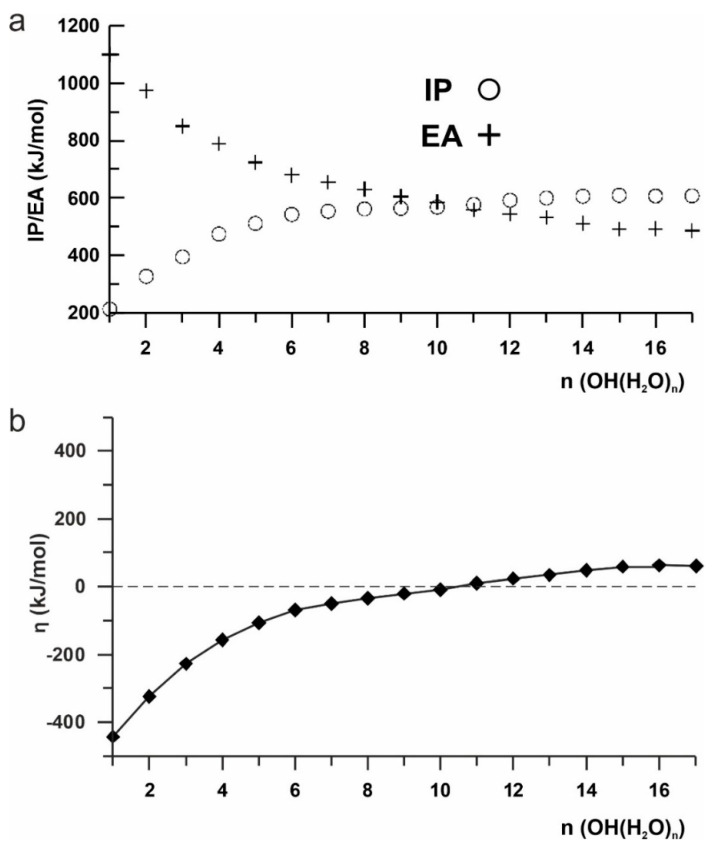
Changes in the values of reactivity parameters: (**a**): ionization potential (IP) and electron affinity (EA); (**b**): global hardness as a function of the [OH(H_2_O)_n_]^−^ cluster size.

## Data Availability

The data that support the findings of this study are available from the author upon reasonable request.

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
