# Peer review of "The Consequences of Water Interactions with Nitrogen-Containing Carbonaceous Quantum Dots—The Mechanistic Studies"

_ijms, 2022, doi:10.3390/ijms232214292_

Round 1

Reviewer 1 Report (Previous Reviewer 1)

In this article the author reports ‘The consequences of water interactions with nitrogen-containing carbonaceous quantum dots – the mechanistic studies’. This topic could be interesting for the readers. However, the paper needs a major revision before publication. I have listed a few comments that need to be addressed:

1.      The novelty of the present article should be discussed in the Introduction section.

2.      Introduction could be much better with more background about the work with up-to-date citations.

3.      Write the full form once at the first appearance.

4.      The authors may provide better HRTEM image of the N-CQDs

5.      The author should write the purpose for each test in one/two sentences (in brief) before explaining the results of the characterization techniques. Therefore, the logic and organization of this part will be enhanced.

6.      Conclusion should be precise with key findings of the results.

7.      Carefully revise the typos and linguistic errors to make the manuscript error-free.

8.      Introduction could be much better with more background about the carbon quantum dots with up-to-date citations. Also, I would advise adding a few recent reviews article on carbon dots in the introduction part such as Nature Nanotechnology, 17(2), 112-130; Beilstein Journal of Nanotechnology, 13(1), 1068-1107; ACS Applied Bio Materials, 3(11), 8023-8031; Coordination Chemistry Reviews 470 (2022): 214703; Nanoscale Advances, 2021, 3, 6762-6796 (DOI: 10.1039/D1NA00447F)

Author Response

Author's Reply to the Review Report (Reviewer 1)

Comments and Suggestions for Authors

In this article the author reports ‘The consequences of water interactions with nitrogen-containing carbonaceous quantum dots – the mechanistic studies’. This topic could be interesting for the readers. However, the paper needs a major revision before publication. I have listed a few comments that need to be addressed:

  1. The novelty of the present article should be discussed in the Introduction section.

---------------------------------------------------------------------------------------------------------------------------------------------

Dear Referee, thank you for this remark, it is corrected now

---------------------------------------------------------------------------------------------------------------------------------------------

  1. Introduction could be much better with more background about the work with up-to-date citations.

---------------------------------------------------------------------------------------------------------------------------------------------

Corrected, please see pt. 8

---------------------------------------------------------------------------------------------------------------------------------------------

  1. Write the full form once at the first appearance.

---------------------------------------------------------------------------------------------------------------------------------------------

Dear Referee, thank you for this remark, it is corrected now

---------------------------------------------------------------------------------------------------------------------------------------------

  1. The authors may provide better HRTEM image of the N-CQDs

---------------------------------------------------------------------------------------------------------------------------------------------

Dear Referee, thank you for this remark, it is corrected now (Fig. S7)

---------------------------------------------------------------------------------------------------------------------------------------------

  1. The author should write the purpose for each test in one/two sentences (in brief) before explaining the results of the characterization techniques. Therefore, the logic and organization of this part will be enhanced.

---------------------------------------------------------------------------------------------------------------------------------------------

Dear Referee, thank you for this remark, it is corrected now

---------------------------------------------------------------------------------------------------------------------------------------------

  1. Conclusion should be precise with key findings of the results.

---------------------------------------------------------------------------------------------------------------------------------------------

Dear Referee, thank you for this remark, it is corrected now

---------------------------------------------------------------------------------------------------------------------------------------------

  1. Carefully revise the typos and linguistic errors to make the manuscript error-free.

---------------------------------------------------------------------------------------------------------------------------------------------

Dear Referee, thank you for this remark, it is corrected now

---------------------------------------------------------------------------------------------------------------------------------------------

  1. Introduction could be much better with more background about the carbon quantum dots with up-to-date citations. Also, I would advise adding a few recent reviews article on carbon dots in the introduction part such as Nature Nanotechnology, 17(2), 112-130; Beilstein Journal of Nanotechnology, 13(1), 1068-1107; ACS Applied Bio Materials, 3(11), 8023-8031; Coordination Chemistry Reviews 470 (2022): 214703; Nanoscale Advances, 2021, 3, 6762-6796 (DOI: 10.1039/D1NA00447F)

---------------------------------------------------------------------------------------------------------------------------------------------

Dear Referee, thank you for this remark, it is corrected now

---------------------------------------------------------------------------------------------------------------------------------------------

Reviewer 2 Report (Previous Reviewer 3)

Author reported clearly studied  the interaction of water with carbon quantum dots.

The interaction between these species and water is of capital relevance for any kind of aplications. 

The physiochemical part of te manuscript is well strucutred and clear but the are some issues and few limitation about this study.

FIrstly, author reported that a fully characterizaiton of the c-dots i described in referece 8 and 22. Authot should consider to include these characterizaiton in teh manuscript or at least in teh supporting information. The reproducibility of each batch of c-dots is not gurantee and if a different batch has been use it must be characterized again. 

The main issue os this study is related to the limited application to N-CQDs. C-dots family is quite vast. Is this study really relevant for teh descriptio of the water -cots interaction? As it is presented it is limited only to N-CQDs. Could authors make some consideration about?

Considering the above mentioned issues, this paper cannot be accepted as it is but it is quite valuable and i request major revisions.

Author Response

Author's Reply to the Review Report (Reviewer 2)

Author reported clearly studied  the interaction of water with carbon quantum dots.

The interaction between these species and water is of capital relevance for any kind of aplications. 

The physiochemical part of te manuscript is well strucutred and clear but the are some issues and few limitation about this study.

FIrstly, author reported that a fully characterizaiton of the c-dots i described in referece 8 and 22. Authot should consider to include these characterizaiton in teh manuscript or at least in teh supporting information. The reproducibility of each batch of c-dots is not gurantee and if a different batch has been use it must be characterized again. 

---------------------------------------------------------------------------------------------------------------------------------------------

Dear Referee, thank you for this remark, I fully agree with you. Our experience confirm your apprehension, that’s why sometimes we prefer “larger synthesis”.  In fact the described here material is from the same batch.

---------------------------------------------------------------------------------------------------------------------------------------------

The main issue os this study is related to the limited application to N-CQDs. C-dots family is quite vast. Is this study really relevant for teh descriptio of the water -cots interaction? As it is presented it is limited only to N-CQDs. Could authors make some consideration about?

---------------------------------------------------------------------------------------------------------------------------------------------

Dear Referee, thank you for this remark, the analysis of 5 different CQDs will appear in near future.

---------------------------------------------------------------------------------------------------------------------------------------------

Considering the above mentioned issues, this paper cannot be accepted as it is but it is quite valuable and i request major revisions.

Round 2

Reviewer 1 Report (Previous Reviewer 1)

The authors have addressed all the questions raised before. Therefore, the manuscript can be accepted in its present form.

Reviewer 2 Report (Previous Reviewer 3)

i endorse this paper without asking for any addiitonal revisions.

This manuscript is a resubmission of an earlier submission. The following is a list of the peer review reports and author responses from that submission.

Round 1

Reviewer 1 Report

In this manuscript the author reports ‘The consequences of water interactions with nitrogen-containing carbonaceous quantum dots – the mechanistic studies’. This work will be surely interesting for many researchers working in the related fields. However, there are some points that need to be addressed before possible publication.

 Recommendation: Major as noted

1.  The novelty of the present article should be discussed little more in the Introduction section.

2.      Did the authors perform any purification steps after hydrothermal reaction? That should be mentioned in the synthesis process.

3.   The formatting and grammatical errors in the article need to be checked carefully.

4.      The author should write the purpose for each test in one/two sentences (in brief) before explaining the results of the characterization techniques. Therefore, the logic and organization of this part will be enhanced.

5.      The authors have cited relevant references in the Introduction section; however the manuscript needs to be highlighted with some recent reports to further broaden the impact, related literatures: Materials, 14(9), 2454; ACS Applied Materials & Interfaces, 12(46), 51940-51951; Carbon 156 (2020): 110-118; Nanoscale Advances (2021) 3, 6762-6796 (DOI: 10.1039/D1NA00447F); Advances in Colloid and Interface Science 259 (2018): 44-64.

Author Response

Reviewer 1

In this manuscript the author reports ‘The consequences of water interactions with nitrogen-containing carbonaceous quantum dots – the mechanistic studies’. This work will be surely interesting for many researchers working in the related fields. However, there are some points that need to be addressed before possible publication.

 Recommendation: Major as noted

  1. The novelty of the present article should be discussed little more in the Introduction section.

Dear Referee, thank you for this remark,

  1. Did the authors perform any purification steps after hydrothermal reaction? That should be mentioned in the synthesis process.

Dear Referee, thank you for this remark; full physicochemical characterization including the effect of additional purification steps was performed and described recently in ref. 19

  1.  The formatting and grammatical errors in the article need to be checked carefully.

Dear Referee, thank you for this remark, I did my best to correct it.

  1. The author should write the purpose for each test in one/two sentences (in brief) before explaining the results of the characterization techniques. Therefore, the logic and organization of this part will be enhanced.

Dear Referee, thank you for this remark; indeed, the organization of R&D section could be mist, the problem is that, this is the template. I tried to put some small sentences as the introduction, the effect is that for other Referees it is too much informal.

  1. The authors have cited relevant references in the Introduction section; however the manuscript needs to be highlighted with some recent reports to further broaden the impact, related literatures: Materials, 14(9), 2454; ACS Applied Materials & Interfaces, 12(46), 51940-51951; Carbon 156 (2020): 110-118; Nanoscale Advances (2021) 3, 6762-6796 (DOI: 10.1039/D1NA00447F); Advances in Colloid and Interface Science 259 (2018): 44-64.

Dear Referee, thank you for this remark, the additional literature was added.

Reviewer 2 Report

This paper by WiÅ›niewski describes mechanistic interactions of carbonaceous quantum dots with water. 

The paper is mostly theoretical, but presents nice outcomes from the research carried out. The language style should be changed as it is very informal and non scientific in some rhetorical considerations and expressions, self questions. 

It needs some improvements before further processing: 

- It states the synthesis of the quantum dots, but no electron microscopy images are presented. TEM is highly appreciated, to know the exact size of the used dots. 

- The same for zeta potential measurements. The surface charge of the quantum dots has great significance in water - and surroundings interactions. 

I suggest major revisions.

Author Response

Reviewer 2

This paper by WiÅ›niewski describes mechanistic interactions of carbonaceous quantum dots with water. 

The paper is mostly theoretical, but presents nice outcomes from the research carried out. The language style should be changed as it is very informal and non scientific in some rhetorical considerations and expressions, self questions. 

Dear Referee, thank you for this remark, but I cannot agree with you fully.

  • The work is mostly experimental, and the theoretical investigations were added to explain the observables
  • The self-questions – corrected;

It needs some improvements before further processing: 

- It states the synthesis of the quantum dots, but no electron microscopy images are presented. TEM is highly appreciated, to know the exact size of the used dots. 

Dear Referee, thank you for this remark. To avoid plagiarism, it wasn't possible to place the full HRTEM analysis - there are references to previous studies 8 & 19; nevertheless, new HRTEM pictures are shown as Fig. S7

- The same for zeta potential measurements. The surface charge of the quantum dots has great significance in water - and surroundings interactions. 

Dear Referee, thank you for this remark, I fully agree with you, such studies are planned for the near future. Unfortunately, this needs funds,   for funds we need papers, and for papers we need results....

I suggest major revisions.

Dear Referee, thank you

Reviewer 3 Report

Authors reported a very interesting paper about a critical issue: the interaction of water with carbon quantum dots.

The paper is limited in the analysis of just one family of carbon dots and this represents the main limitation of the work. Nevertheless, the data reported,  even if they are limeted, firtly and clearly enlithen the iteraction of water with teh outer shel of carbon dots.

The proves reported here could let teh foundation of new discoveris in structure and interaction of c-dots with plenty of substrates in biological environement.

There few issue in teh manuscript such teh use of acronyms in the abstract (Please do not use any acronyms in the abstract!) and some typos (line 247: citation missed). And plase check the grammar and make more clear and readable the methodological section.

Nonetheless, this paper is fairly up to mark for publication and i warmly endorse it.

Author Response

please find the response in the attached file

Round 2

Reviewer 1 Report

The authors have addressed all the questions raised before. Therefore the manuscript can be accepted in the present form.

Author Response

attached file

Reviewer 2 Report

The authors have provided a revised version of the manuscript. However, the way they addressed the comments, answers and implemented changes is not satisfactory. 

Electron microscopy images do not support what is stated in the text. Nor TEM or SEM micrographs show clearly de quantum dots particles. Size distribution histogram is missing.  In addition, authors must not justify the missing data saying that they need funds to cover the experiments (current and future) and that for that they need papers... this is not acceptable. 

Author Response

please find in attachment
